# On the Pitfalls of Visual Learning in Referential Games

**Shresth Verma**
ABV-Indian Institute of Information Technology and Management
Gwalior, India
vermashresth@gmail.com

## Abstract

This paper focuses on the effect of game design and visual representations of real-world entities on emergent languages in referential games. Strikingly, we find that the agents in such games learn to successfully communicate even when provided with visual features from a randomly initialized neural network. Through a series of experiments, we highlight the agents' inability to effectively utilize high-level features. Using Gradient weighted-Class Activation Mapping, we verify that the agents often 'look' at regions not related to entities. Culminating with a positive result, we show how environmental pressure from agent population can nudge the learners into effectively capturing high-level visual features.

## 1  Introduction

The ability to communicate through natural language is often seen as a long standing goal in AI. The current prevalent paradigm in natural language learning leverages statistical patterns using large amounts of data. Alternatively, the field of language emergence [2, 9, 29, 13] exploits language's functional nature. Recently, there has been a revival of interest in language emergence games by studying them through the lens of neural agents [20, 14, 21]. Efforts are made to understand the linguistic properties of emergent languages, such as grammar [30], compositionality [1], priors and biases [6, 11] and capacity and bandwidth [24]. Another line of work stresses upon the shortcomings of emergent language through neural agents, such as anti-efficient encoding [5] and the need for specialized environmental pressures [18, 25]. However, an area that is often ignored is the effect of visual representations on shaping the emergent language properties. [3] demonstrate an alarming behaviour in emergent languages where agents rely on low-level visual features for communication. In our work, we highlight the role of visual representations and game design in aggravating this behaviour. Through empirical results, we show that the agents are ineffective at learning from high-level visual representations of images. This is surprising since it has been common practice in emergent language literature to represent real-world images as either final layer activations of pre-trained deep ConvNets such as VGG-16 [28] (e.g. in [14, 20, 19, 12]) or using representations learned in-game [8, 25, 22]. Further, we reformulate Gradient-weighted Class Activation Mapping (Grad-CAM) [26] to visualize where the agents are looking. This is the first application of Grad-CAM in interpreting referential games. Lastly, we conduct experiments in line with strong results obtained from language emergence in population of agents [17, 16, 23, 7]. We show that a simple population-based Referential Game induces a regularization effect, nudging the agents to communicate through high-level visual features. Our novel contributions are thus as follows:

1) We evaluate agents' reliance on low-level features for communication using two synthetic testbeds in Emergent Communication.

2) We reformulate Grad-CAM for use in Multi-Agent Reinforcement Learning setting and show that the agents 'look' at regions not related to entities.

36th Conference on Neural Information Processing Systems (NeurIPS 2022).

3) We demonstrate that environmental pressure through population of agents regularize agents to use high-level visual features for communication.

## 2 Referential Game

Two agents, a sender $S$ and a receiver $R$ play the game. The sender is shown a target image $t$ and $K$ distracting images $\mathcal{D} = \{d_k\}_{k=1}^{i=K}$. Consequently, sender must come up with a message $m$ of length $l$ with tokens from vocabulary $\mathcal{V}$. The receiver is shown the target image and distracting images in a shuffled order, along with the message $m$. A payoff of 1 is provided to both the agents iff the receiver identifies the target correctly.

We consider the following agent and game variants:

**Distractor Aware Sender**: Given the target image $t$ and distracting images $\mathcal{D}$, sender must come up with message $m$.

**Distractor Agnostic Sender**: Given only target image $t$, sender must come up with message $m$.

**Same Target Game**: Given message $m$ and a shuffled set of images containing $t$ and $\mathcal{D}$, the receiver must identify the target correctly.

**Different Target Game**: Instead of target image $t$, the receiver is shown a different target image $t'$, which is conceptually same as $t$.

We use $K = 1, l = 5, |\mathcal{V}| = 10$ in all experiments. Previous works have shown that even a small game size can be a challenging learning environment. Crucially, keeping these hyperparameters fixed makes sure that there are no confounding factors when studying the effects of game variations.

### 2.1 Data

We generate two synthetic datasets [1] , SHAPES and BULLET, to represent images corresponding to different visual attributes. Both datasets consist of 30 X 30 images with objects characterized by one of 3 colours, 2 sizes and 3 shapes. SHAPES has 2D objects placed on a background with randomized location. BULLET has 3D objects placed on a checkered plane with randomized location as well as camera viewpoint. Unlike SHAPES where background is constant, BULLET has a checkered plane where pixel values change with changing camera orientation. Thus, BULLET showcases more variation in pixel-level features as compared to SHAPES, while having the same high-level attribute set. We argue that studying emergent languages using proposed datasets has an advantage over ImageNet [10], as attribute information can readily be controlled. Further, raw pixel inputs provide a much more realistic task [19], than using one-hot vectors of attributes, as has been done in many prior works [18, 4].

### 2.2 Learning and Architecture

We re-implement agent architectures from [14]. The sender and receiver both have one ConvNet visual module $f_s$ and $f_r$ and one LSTM language module $L_s$ and $L_r$ respectively. The visual module maps input images to fixed dimension embedding. Sender's language module $L_s$ generates sequence of message tokens $m$ with embeddings from $f_s$ as the initial hidden state. The receiver's language module $L_r$ receives message $m$ and a message embedding is obtained by performing affine transformation $g$ on last hidden state in $L_r$. Finally, the receiver compares the message embedding and visual embeddings from each of the images to point to a target. We use Straight-Through Gumbel-softmax (GS) relaxation [15], while sampling messages from sender's language module. This allows the game to be end-to-end differentiable which is crucial for Grad-CAM (Section 4.2). The energy function for each image $v$, $E(v, m) = f_r(v)^T g(L_r(m))$, can be used to define probability distribution over images as $p(v|m) \propto e^{E(v,m)}$. Since our learning setup is similar to the one proposed by [14], we optimize the same loss function. $\quad \mathcal{L} = \mathbb{E}_{m \sim \Pi_\theta} \sum_{k=1}^{K-1} max(0, 1 - E(t, m)) + E(d_k, m)$

To study the effect of visual representations, we also include the following two learning variants:

**In-game learning (I)**: The weights for both $f_r$ and $f_s$ are updated during learning in game play.

**Pre-trained classifier (P)**: A visual module is pre-trained on the task of classifying entities. Weights are saved and representations of input images are computed before gameplay. $f_s$ and $f_r$ are not used.

---

[1]https://github.com/vermashresth/Referential_Shapes

| Metric | Distractor Aware | | | | | | Distractor Agnostic | | | | | |
|---|---|---|---|---|---|---|---|---|---|---|---|---|
| | Same Target | | | Diff Target | | | Same Target | | | Diff Target | | |
| | I | P | R | I | P | R | I | P | R | I | P | R |
| **SHAPES** | | | | | | | | | | | | |
| Comm Acc | 98.3∓0.72 | 98.03∓0.9 | 98.5∓0.5 | 96.57∓1.9 | 95.8∓1.7 | 95.4∓1.8 | 94.32∓1.8 | 94.04∓2.0 | 94.02∓1.8 | 88.16∓5.1 | 86.5∓4.0 | 86.39∓3.0 |
| Noise Acc | 87.94∓8.8 | 63.5∓4.4 | 71.24∓8.1 | 56.25∓7.9 | 56.6∓8.1 | 57.2∓4.8 | 55.9∓2.7 | 59.1∓9.5 | 51.02∓8.1 | 50.26∓2.2 | 51.56∓3.3 | 53.9∓9.2 |
| General Acc | 93.2∓0.94 | 92.8∓1.4 | 96.93∓1.2 | 72.7∓0.80 | 78.3∓0.91 | 96.93∓1.2 | 66.18∓0.69 | 67.3∓1.4 | 86.7∓0.93 | 68.2∓0.71 | 69.3∓1.0 | 86.7∓0.93 |
| Top Sim | 0.07∓0.14 | 0.03∓0.13 | 0.02∓0.13 | 0.05∓0.10 | 0.01∓0.13 | 0.08∓0.15 | 0.28∓0.11 | 0.14∓0.14 | 0.12∓0.14 | 0.41∓0.11 | 0.20∓0.12 | 0.38∓0.12 |
| **BULLET** | | | | | | | | | | | | |
| Comm Acc | 98.25∓0.76 | 97.77∓1.1 | 97.27∓0.83 | 96.59∓1.6 | 96.16∓1.7 | 95.45∓1.7 | 94.98∓1.9 | 94.2∓2.0 | 93.71∓2.2 | 88.93∓4.1 | 87.18∓3.4 | 87.78∓2.4 |
| Noise Acc | 86.86∓7.8 | 86.41∓1.21 | 84.05∓4.1 | 63∓2.5 | 62.8∓3.9 | 60.71∓2.6 | 51.49∓1.3 | 61.22∓5.6 | 65.96∓3.6 | 50.78∓2.0 | 51.68∓1.5 | 52.2∓2.7 |
| General Acc | 95.3∓0.96 | 94.11∓0.6 | 96.7∓0.9 | 92.6∓0.94 | 90.1∓1.8 | 96.7∓0.9 | 82.2∓0.83 | 80.8∓0.77 | 81.5∓0.9 | 69.6∓0.71 | 66.8∓0.9 | 81.5∓0.9 |
| Top Sim | 0.06∓0.15 | 0.019∓0.14 | 0.045∓0.13 | 0.08∓0.15 | 0.09∓0.11 | 0.08∓0.14 | 0.29∓0.15 | 0.25∓0.15 | 0.19∓0.17 | 0.32∓0.13 | 0.29∓0.15 | 0.33∓0.13 |

Table 1: Evaluation metrics (mean and std-dev) on different variations of referential game.

# 3 Evaluation Metrics

We evaluate the game variants on four metrics: **(1) Communication Success:** Measured as average reward obtained at convergence. **(2) Generalization Success:** Measured as communication success on novel combinations of unseen concepts. **(3) Topographical Similarity:** Measured as correlation between all-pair distances in message space and in input space and can be seen as language compositionality measure [20]. **(4) Noise Accuracy:** Measured as test time communication success over images containing only gaussian noise. It quantifies agents' reliance on pixel-level visual features [3].

# 4 Experiments

## 4.1 Effects from game variations

All combinations of proposed game variants are enumerated and run until convergence for 10 different seed values. Evaluation metrics are reported in Table 1. To develop a baseline for comparison, we also report results from games with a **random visual module (R)**. In this setting, $f_s$ and $f_r$ are fully connected single layer networks with weights randomly initialized and frozen during training.

We obtain the following statistically significant observations from our experiments:

- High levels of communication success are reached in all the games ($> 90\%$). Surprisingly, experiments with *Random Visual Module* also achieve high communication success.
- Noise accuracy is lower in *Distractor-Agnostic* setting as compared to *Distractor-Aware*. A plausible reason could be the absence of context for sender without distractors. Since relative comparison is not possible, encoding high-level visual information would be incentivized. Higher topographical similarity in *Distractor-Agnostic* also supports our argument.
- While *Distractor-Aware* setting has a high noise accuracy, using *Different-Target* lowers it as compared to *Same-Target* game. This matches with the results obtained in [3].
- All *Distractor-Aware Pre-Train* experiments have performance comparable to baseline (random) experiment. This is an alarming result as it suggests that agents gain no significant benefit in communication when provided with high-level visual information.

Statistical significance is calculated using t-test for comparison of means when comparing two groups. All the above observations are significance with p-value $< 0.05$.

## 4.2 Where are the agents looking?

To qualitatively access whether the agents are looking at high-level visual attributes or at pixel-level differences, we reformulate Gradient-weighted Class Activation Mapping [26]. As the game is end-to-end differentiable, we can attribute the outcome of the game to specific regions in the inputs by generating a class discriminative localization map. Let $m$ be the message generated by sender and $u \in t \cup \mathcal{D}$ be the image chosen by the receiver. The energy function $E(u, m)$ represents the likelihood of $u$ being the target image. For any image $v \in t \cup \mathcal{D}$, for which we want to generate localization map $H$, we define $A_s^u$ and $A_r^u$ as activations from final convolutional layers in sender's and receiver's visual modules. We calculate the localization map for sender as $\alpha_s^u = \frac{\partial E(v,m)}{\partial A_s^u}, H_s^u = ReLU(\alpha_s^u A_s^u)$

We repeat this process for receiver and for all the images in the sender's and receiver's input. Figure 1 shows that grad-CAM can accurately highlight regions of entities in targets and distractors. However,

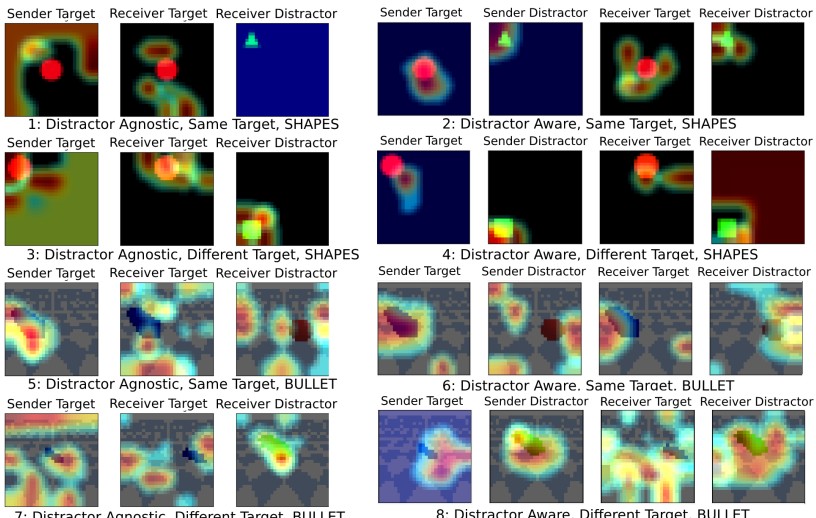

Figure 1: Grad-CAM heatmaps for different referential game variants (all with Pre-trained (P) visual module), averaged over 5 seeded runs.

in many cases, the agents "look" at background regions. This is more prominent in BULLET dataset. Interestingly, in some cases, agents look at negative space of the object. This suggests that agents may communicate through negation of attributes (e.g. 'the object not in the bottom right corner') as has been previously expressed in [8, 27].

### 4.3 Population based Referential Game

We extend the referential game setup to a population of agents. Consider $N$ agents, consisting of an equal number of senders and receivers. At every game turn, a referential game is played between a random pair of sender and a receiver. We argue that population as an environmental pressure should reduce the agents' reliance on low-level features. This is because high communication success from low-level features is a sign of conceptual pact between two agents. However, when the game partners are continuously shuffled, low-level pacts would no longer always match. Figure 2 shows that noise accuracy at convergence indeed reduces with larger population sizes.

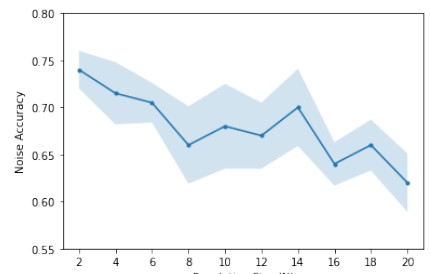

Figure 2: Noise Accuracy at convergence for different population sizes. *Distractor Aware*, *Same Target* game on BULLET dataset

## 5 Conclusion

Our work ascertains that agents in referential games often rely on low-level features for communication; however, some game settings are more susceptible to do so than others. Thus, noise accuracy should always be reported in emergent language experiments. Moreover, a high generalization accuracy can be misleading as it may result from pixel-level comparison rather than a compositional language. Additionally, as we observe a prevalence of negation of attributes in emergent communication, there is a need to incorporate it in compositionality measures.

The multi-input, multi-output Grad-CAM setup through GS channel, presented in our work, can be adapted as an interpretability tool for numerous Multi-Agent RL scenarios such as visual dialog agents, embodied agents or in cooperative/competitive communication games. Lastly, a promising direction for future work is the effect of population interaction on conceptual pacts among agents and how it evolves over time.

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
