# OpenReview forum: "On the Pitfalls of Visual Learning in Referential Games"
_NeurIPS.cc/2022/Workshop/LaReL — LaReL 2022_

### Official Review · Reviewer_uXAC · 2022-10-05
**Insightful work**

**Rating:** 7
**Confidence:** 3

**Review:**

This is a nice short work with plenty of experiments and analyses. The authors investigate visual emergent communication systems, specifically on the aspects of 1) what features these systems rely on, and 2) are these features generalizable.

Among the proposed methods, I especially like the visualization produced by Grad-CAM. I agree with the authors that this method can potentially be used in many other tasks to help better understanding model/agent behaviours, for example, to help detecting spurious correlations in vision tasks with photos.

On a related note, from Figure 1 I find that although as the authors describe, "the agents look at background regions", their attentions often somewhat cover the edges of the object, this might have provided the models sufficient information to distinguish between distractors?

Another concern is about the complexity of the two tasks. In such tasks (specifically, single object, same background), are higher level features really needed? It would be very helpful to see the same set of experiments on some slightly more complex settings, do the observations still hold.

Overall I like this work, I think it is quite insightful, the method can be extended to many other tasks as a way to boost explainability.

---

### Official Review · Reviewer_ikyo · 2022-10-17
**Interesting experimental investigation**

**Rating:** 7
**Confidence:** 4

**Review:**

This work proposes to study what are the key visual features on which agents agree during referential games. More specifically it shows that agents use low-level features to communicate about visual referents.

Through a series of thorough experiments with different referential game settings, the authors first observe that even though agents reach high game accuracy, they display high noise accuracy indicating that agents rely on low-level features. The fact that using pre-trained representation does not improve performance seems to corroborate this hypothesis.

In addition to this benchmark, the authors propose to visually analyze the pixels of interest during referential games using Grad-CAM. This investigation is original and interesting. However, I do not really know what to extract from the analysis. The authors claim that the agent only focuses on the background ("reasoning with a negation"). But extracting the shape of an object in negative seems to me to be a manifestation of high-level feature exploitation. Moreover, the images provided in Figure 1 seem to illustrate that agents focus on the edges of objects. Providing metrics in the image space could help refine this visual analysis.

Overall, I think that this paper is a good fit for this workshop.

---

### Decision · Program_Chairs · 2022-10-20

Accept